# Effects of Peptide Receptor Radiotherapy in Patients with Advanced Paraganglioma and Pheochromocytoma: A Nation-Wide Cohort Study

**DOI:** 10.3390/cancers16071349

**Published:** 2024-03-29

**Authors:** Linda Skibsted Kornerup, Mikkel Andreassen, Ulrich Knigge, Anne Kirstine Arveschoug, Per Løgstup Poulsen, Andreas Kjær, Peter Sandor Oturai, Henning Grønbæk, Gitte Dam

**Affiliations:** 1Department of Hepatology & Gastroenterology, ENETS Center of Excellence, Aarhus University Hospital, 8200 Aarhus, Denmark; lindajen@rm.dk (L.S.K.); henngroe@rm.dk (H.G.); gittedam@rm.dk (G.D.); 2Department of Endocrinology, ENETS Center of Excellence, Rigshospitalet, 2100 Copenhagen, Denmark; ulrich.knigge@regionh.dk; 3Department of Surgery and Transplantation, ENETS Center of Excellence, Rigshospitalet, 2100 Copenhagen, Denmark; 4Department of Nuclear Medicine & PET, ENETS Center of Excellence, Aarhus University Hospital, 8200 Aarhus, Denmark; annearve@rm.dk; 5Department of Endocrinology, ENETS Center of Excellence, Aarhus University Hospital, 8200 Aarhus, Denmark; perpouls@rm.dk; 6Department of Clinical Physiology and Nuclear Medicine, ENETS Center of Excellence, Rigshospitalet, 2100 Copenhagen, Denmark; akjaer@sund.ku.dk (A.K.); peter.sandor.oturai@regionh.dk (P.S.O.); 7Department of Clinical Medicine, Aarhus University, 8200 Aarhus, Denmark

**Keywords:** pheochromocytoma, paraganglioma, malignant, PRRT

## Abstract

**Simple Summary:**

Pheochromocytomas and paragangliomas are rare neuroendocrine tumours that originate from the adrenal medulla or extra-adrenal sympathetic or parasympathetic ganglia. They have the potential to secrete catecholamines, leading to e.g., hypertension. The only curative treatment modality is radical surgery. In disseminated diseases, limited data are available to guide proper medical treatment. Peptide receptor radionuclide therapy (PRRT) represents a rather novel treatment modality for these tumours. In the present nation-wide study, including all patients (n = 28) treated with PPRT in Denmark, we show a median overall survival of 72 and a median progression-free survival of 30 months with low toxicity. In conclusion, PRRT seems to represent an effective treatment option. The sequence of PRRT in treatment algorithms for pheochromocytomas and paragangliomas remains to be clarified.

**Abstract:**

Introduction: Pheochromocytomas and paragangliomas are rare neuroendocrine tumours that originate from chromaffin cells within the adrenal medulla or extra-adrenal sympathetic ganglia. Management of disseminated or metastatic pheochromocytomas and paragangliomas continues to pose challenges and relies on limited evidence. Method: In this study, we report retrospective data on median overall survival (OS) and median progression-free survival (PFS) for all Danish patients treated with peptide receptor radionuclide therapy (PRRT) with ^177^Lu-Dotatate or ^90^Y-Dotatate over the past 15 years. One standard treatment of PRRT consisted of 4 consecutive cycles with 8–14-week intervals. Results: We included 28 patients; 10 were diagnosed with pheochromocytoma and 18 with paraganglioma. Median age at first PRRT was 47 (IQR 15–76) years. The median follow-up time was 31 (IQR 17–37) months. Eight patients died during follow-up. Median OS was 72 months, and 5-year survival was 65% with no difference between pheochromocytoma and paraganglioma. Patients with germline mutations had better survival than patients without mutations (*p* = 0.041). Median PFS after the first cycle of PRRT was 30 months. For patients who previously received systemic treatment, the median PFS was 19 months, compared with 32 months for patients with no previous systemic treatment (*p* = 0.083). Conclusions: The median OS of around 6 years and median PFS of around 2.5 years found in this study are comparable to those reported in previous studies employing PRRT. Based on historical data, the efficacy of PRRT may be superior to ^131^I-MIBG therapy, and targeted therapy with sunitinib and PRRT might therefore be considered as first-line treatment in this patient group.

## 1. Introduction

Pheochromocytomas and paragangliomas are rare neuroendocrine tumours that originate from chromaffin cells within the adrenal medulla or extra-adrenal sympathetic ganglia, respectively. These tumours have the potential to secrete catecholamines, leading to a range of clinical symptoms, including hypertension, palpitations, and paroxysmal attacks [1,2]. Approximately 10% of pheochromocytomas and 40% of paragangliomas are classified as malignant. Germline genetic mutations have been identified in roughly 30% of patients. Certain mutations, especially those affecting the SDHx genes, are associated with a higher probability of malignant transformation [3,4]. Due to the high prevalence of germline mutations, guidelines recommend genetic testing in all patients diagnosed with pheochromocytoma or paraganglioma [5].

While surgical resection remains the primary treatment for localized disease [6], the management of disseminated or metastatic pheochromocytomas and paragangliomas continues to pose challenges and relies on limited evidence. Treatment options for disseminated disease encompass conventional chemotherapeutic agents such as cyclophosphamide, vincristine, and dacarbazine [3]. In addition to chemotherapy, targeted treatment with the tyrosine kinase inhibitor (TKI), sunitinib, may be used. This is supported by observational studies and one randomized clinical trial, demonstrating a progression-free survival benefit of 5 months compared to a placebo [7,8].

Targeted radionuclide therapies (TRTs) represent an alternative with fewer side effects compared to chemotherapy. One such therapy utilized for over four decades is the ^131^I-Meta-Iodobenzylguanidine (MIBG) treatment, which targets the norepinephrine transporter on pheochromocytoma and paraganglioma cells, delivering radiation directly to the tumour cells [9]. Peptide receptor radiotherapy (PRRT) with ^177^Lu- or ^90^Y-Dota-peptides represents a more recently developed TRT that exploits the overexpression of somatostatin receptors (SSTR) on pheochromocytoma and paraganglioma cells [10]. Dotatate and Dotatoc are somatostatin analogues that, labelled with a beta-emitting radioisotope, selectively deliver radiation to the tumour cells. PRRT has primarily been employed for well-differentiated neuroendocrine gastroenteropancreatic tumours [10,11].

Limited data are available on the efficacy of PRRT in pheochromocytomas and paragangliomas. A meta-analysis from 2022 of primarily retrospective studies suggested superior efficacy of PRRT compared to ^131^I-MIBG, with a ten-month extension in progression-free survival [12]. However, these studies display considerable heterogeneity. For patients suitable for both ^131^I-MIBG and PRRT based on tracer uptake, it remains unclear which should be the first-line therapy [13]. In this study, we report data on median overall survival (OS) and median progression-free survival (PFS) for all Danish patients treated with PRRT over the past 15 years.

## 2. Methods

### 2.1. Patients

In Denmark, PRRT exclusively takes place at Rigshospitalet, Copenhagen (patients from East Denmark), and Aarhus University Hospital, Aarhus (patients from West Denmark), both ENETS Centres of Excellence.

In this retrospective cohort study, we included all Danish patients diagnosed with pheochromocytomas and paragangliomas who have ever received PRRT. Demographics, clinical, biochemical, histopathological, and imaging data were obtained from electronic patient records. Data were collected from 1 January 2009 to 31 December 2023. 

### 2.2. PRRT

Indications and eligibility criteria for PRRT were evaluated in a multidisciplinary team (MTD) setting. PRRT eligibility included somatostatin receptor imaging (SRI) (^64^Cu-DOTATATE PET/CT or ^68^Ga-DOTATOC PET/CT) showing distribution of tracer in primary tumour and/or metastases with activity higher than in normal liver tissue (Krenning score ≥ 3); sufficient bone marrow function (thrombocytes > 100 × 10^9^/L, white blood cells > 2 × 10^9^ and haemoglobin > 6 mmol/L); standard GFR > 50 mL/min/1.73 m^2^ (measured by ^99m^Tc-DTPA; diethylene-triamine-pentaacetate). All patients had a WHO performance status of 0–2 and were required to be self-sufficient for 24 h during treatment and isolation due to the radiation precautions. The typical length of admission was 2 days.

One standard treatment of PRRT consisted of 4 consecutive cycles with 8–14-week intervals. Between cycles, kidney and bone marrow function were monitored for toxicity. Three months after the last PRRT cycle, a CT scan was conducted, and subsequent imaging was performed at 3–6-month intervals utilizing either CT or SRI in order to monitor treatment response. In cases of progression post-PRRT, patients could undergo another or several treatments of PRRT according to the above-stated criteria.

### 2.3. Outcome

Primary outcomes were median OS and median PFS for PRRT. Median OS was calculated from the first PRRT cycle until death. Median PFS was also calculated from 1st PRRT cycle, until progression defined as radiological or clinical progression assessed through routine clinical practice or death of any course.

### 2.4. Ethics

For the Copenhagen cohort, the study was approved by the Danish Patient Safety Authority (31-1521-453) and by the local data protection agency at Rigshospitalet (20 October 2020). For the Aarhus cohort, the study was approved as a quality assurance project by The Central Denmark Region Committees on Health Research Ethics (ref no 681677 and case no. 1-16-02-120-20). 

### 2.5. Statistics

Continuous variables are presented as medians with range unless otherwise noted. Categorical data are shown as absolute numbers (valid cases, percentages). The chi-square test and two-sample Wilcoxon test were used to test the distribution of binomial and continuous data as appropriate.

We used the Kaplan–Meier function to compute survival probabilities. All values of OS and PFS are reported as the median. Log-rank and univariate Cox regression tests were used to test associations. The following potential risk factors were investigated in univariate analysis; gender, age, germline mutation and previous systemic treatment. Tests were considered statistically significant when *p*-values < 0.05 (two-sided test). Statistical analyses were performed using STATA version 13.1 for Mac (StataCorp LP, College Station, TX, USA).

## 3. Results

We included 28 patients diagnosed with pheochromocytomas or paragangliomas having received PRRT. Treatments were administered from November 2009 to October 2022. A total of 10 patients were diagnosed with pheochromocytoma (36%), and 18 were diagnosed with paraganglioma. Of them, 61% were men, and the median age at 1st PRRT cycle was 47 (15–75) years, see Table 1. Seventeen patients had catecholamine-secreting tumours (61%). Previous treatments included surgery (21%), chemotherapy (11%), tyrosine kinase inhibitor treatment (18%), ^131^I-MIBG treatment (25%), and external beam radiotherapy (18%). In total 14 (50%) had PRRT as first line medical treatment.

Germline mutations were diagnosed in 13 patients (46%), with 9 of them having SDHx mutations, 3 having RET mutations, and 1 having NF1 mutation (for further details see Table 2).

All patients underwent one treatment of PRRT. Due to severe bone marrow suppression, one patient only received one of the four scheduled cycles, and another patient only received two cycles, as they previously received two cycles at another European treatment centre before PRRT became available in Denmark. Five patients underwent two treatments, and one patient underwent three treatments (for further details, see Table 2). Further, 2 patients were treated with ^90^Y-DOTATOC, while 26 patients (93%) were treated with ^177^Lu-DOTATOC/^177^Lu-DOTATATE. The mean administered activity per ^177^Lu-DOTATOC/^177^Lu-DOTATATE treatment was 7.6 GBq. One of the two patients treated with ^90^Y-DOTATOC had two treatments with 7.5 GBq each, while the other had four treatments with 3.6 GBq each.

### Overall Survival and Progression-Free Survival

The median follow-up time was 31 months (IQR 17–37). Eight patients (29%) died during follow-up (Figure 1). Median OS was 72 months, and 5-year survival was 65% with no difference between pheochromocytoma and paraganglioma. Patients with germline mutations had significantly better survival probability compared with patients without germline mutations (*p* = 0.041), see Figure 2. Median OS was not reached for patients with germline mutations, while median OS was 35 months for patients without mutations. There was a tendency towards worse survival probability in patients who previously received systemic treatment compared to those without previous systemic treatment (*p* = 0.077). Age (*p* = 0.93), gender (*p* = 0.59) and catecholamine hypersecretion (*p* = 0.36) had no effect on survival.

Median PFS after the first series of PRRT was 30 months, 19 months for paragangliomas and 32 months for pheochromocytomas (*p* = 0.86); see Figure 3. For patients who previously received systemic treatment, the median PFS was 19 months compared with 32 months for patients with no previous systemic treatment (*p* = 0.083). Median PFS in patients with germline mutations was 34 months and 19 months in patients with no mutation (*p* = 0.083). When stratifying by the type of mutation, PFS for patients with SDHB mutations was 18 months, while it was not reached for patients with other mutations (SDHD, RET and NF1).

## 4. Discussion

In this retrospective study, we present the outcome of all patients with pheochromocytomas and paragangliomas treated with PRRT in Denmark from 2009 to 2022 with at least 6 months of follow-up. Median overall survival after PRRT was 72 months with a more favourable prognosis observed in patients with a germline mutation. The median PFS was 31 months with a tendency towards better prognosis in patients naïve to systemic treatment.

Malignant pheochromocytomas and paragangliomas are rare and complex neuroendocrine tumours, and the patients display a certain degree of heterogeneity regarding tumour burden, tumour multiplicity, germline mutations, and previous treatments. Only a small number of studies reporting the outcome of PRRT have been published to date, and these studies are limited by the low number of patients. The present study is the third largest published study reporting nation-wide data on prognosis after PRRT in this rare disease entity [14,15].

The results on OS align with most previous studies. A study by Vyakaranam et al. publishing data on 22 Swedish patients found a median OS of around 50 months [16], while Kolasinska-Cwikła et al. reported a median OS of 68 months after ^90^Y-DOTATOC in 13 Polish patients all with SDHx mutations [17]. Zandee et al. reported a median OS of 59 months for patients with sympathetic paragangliomas, while the median OS was not reached for patients with parasympathetic paragangliomas [14]. A recently published study by Severi et al. [15] is one of the very few prospective studies and reported data on 46 patients collected through different experimental phase II protocols, in which 12 patients were treated with ^90^Y-DOTATOC and 34 patients with ^177^Lu-DOTATATE. This study shows a substantially better outcome compared to previous studies, with a median OS of 143 months for ^177^Lu-DOTATATE and 92 months for ^90^Y-DOTATOC. The median PFS was not reached, despite a long median follow-up of 76 months. Another prospective, multicentre study on the efficacy and safety of ^177^Lu-DOTATATE included 11 patients with pheochromocytoma and 28 patients with paraganglioma and reported mean survivals of 32 and 82 months, respectively [18].

The median PFS after treatment with ^177^Lu-DOTATATE was calculated in a review and meta-analysis from 2022 (n = 143) and yielded results similar to ours with a median PFS of 30 months [12]. However, in contrast to our findings, the meta-analysis reported better outcomes in pheochromocytomas. It is worth noting that the study by Severi et al. was not included in the meta-analysis. Another review and meta-analysis of the efficacy and safety of PRRT in these patients reported a pooled mPFS of 37 months and median OS of 55 months but with a high degree of heterogeneity between included studies (total patient number 201, range 5–39) [19]. Most of the patients were treated with ^90^Y-DOTATOC.

The variability in OS and PFS in previous studies can be attributed to several factors, including the limited number of patients, possible differences both in selection criteria for PRRT in individual centres and over time, and differences in the sequence of various treatment modalities. Further, the majority of the studies are retrospective [20,21,22,23,24,25]. Collectively, PRRT is considered both a safe and efficient treatment in patients with metastatic and inoperable pheochromocytomas and paragangliomas [20,26], which is in line with our results, with only 1 patient not completing the full treatment due to severe bone marrow suppression.

In the present study, the survival probability in patients with germline mutations was superior to that of patients without germline mutations. This superiority in survival compared with patients without germline mutations has not been described in previous studies, and it was a somewhat unexpected result given that tumours with, e.g., SDHB mutations are known to have more aggressive phenotypes with the potential for metastases, tumour multiplicity, and recurrence [27,28]. However, it is important to recognise that the group of patients with germline mutations was heterogeneous: 7 had SDHB mutation, 2 had SDHD mutation, 3 had RET mutation, and 1 had NF1 mutation. Although the number of patients is too small for formal statistical analyses, we found that mPFS for SDHB-mutated patients was 18 months, which was substantially shorter than in the total cohort. Thus, our results give some support for an unfavourable outcome in SDHB-mutated patients. The prospective study by Kolasinska-Cwikła et al. included 13 patients with SDHx mutations [17]. They showed that patients with SDHB mutations had significantly worse OS and PFS than patients with SDHD mutations. Median PFS for patients with SDHB mutations in our study was slightly better than in the study of Kolasinska-Cwikła et al. (18 months vs. 12 months, respectively), while mPFS for patients with SDHD mutations was not reached in either of the studies. The patients in the study of Kolasinska-Cwikła et al. were exclusively treated with ^90^Y-DOTATOC and received a lower cumulative radiation dose (32 cycles of PRRT in 13 patients) compared to our patients, who were all treated with ^177^Lu-DOTATATE. Pinato et al. reported PFS of 17 months in 5 patients with SDHB mutations [24], whereas Zovato et al. described partial response or stable disease in 4 patients with SDHD mutations [22]. Whether these differences reflect variations in treatment response or an underlying more aggressive behaviour in SDHB-mutated patients remains unknown. The somatostatin-2 receptor is heterogeneously expressed in paragangliomas and pheochromocytomas, especially in those with SDHx mutations [29,30], and we speculate that this may influence the treatment response to PRRT.

Previous systemic treatment trended to be a negative prognostic factor in our study. This covers all previous treatments with chemotherapy, ^131^I-MIBG and TKI, and this result most likely indicates a group of patients with longer disease duration, more aggressive tumours, higher tumour burden, tumour multiplicity, etc.

Conventional ^131^I-MIBG therapy has been used for decades, and its effectiveness is supported by retrospective studies. A meta-analysis from 2014 exploring various treatment outcomes of conventional ^131^I-MIBG reported 5-year survival rates between 45 and 64% and mPFS ranging from 23 to 29 months [31]. In recent years, high-specific activity ^131^I-MIBG has been developed and has finally gained approval in the US, based on results from a prospective phase II study involving 68 patients with disseminated pheochromocytoma or paraganglioma [32]. In this study, the median OS was 37 months. Notably, this cohort was comparable to ours in terms of age, but unlike our cohort, it primarily consisted of patients with pheochromocytomas. Additionally, the patients treated with high-specific activity ^131^I-MIBG had a history of extensive prior treatments. Based on historical data, PFS appears to be longer after PRRT compared to ^131^I-MIBG. However, since there is no randomized controlled trial (RCT) comparing PRRT with ^131^I-MIBG, the observed differences could be attributed to variations in patient selection. Therefore, based on current knowledge, it remains unclear which of these treatments should be recommended as first-line therapy.

Our study has several strengths and limitations. An important strength is the nation-wide design and the inclusion of all patients with pheochromocytomas or paragangliomas ever treated with PRRT in Denmark, without any loss of follow-up. Notably, both centres follow the same rigorous protocol for follow-up, with stringent intervals for scans and outpatient visits. Furthermore, our cohort is thoroughly characterised regarding demographics, laboratory analyses, and histopathology, and all patients have been tested for germline mutations. However, due to the rarity of paragangliomas and pheochromocytomas, the number of patients is an important limitation for our study as well as for all other studies in this field. Consequently, our results must be interpreted with caution, especially when comparing sub-groups. The retrospective study design inherently faces the risk of bias, such as selection bias. Additionally, due to the low number of patients, we were not able to utilize multivariable analyses in an attempt to identify independent risk factors for the outcomes. Another limitation of our study is the rather short follow-up of 2.5 years.

In conclusion, in this nation-wide study, PRRT represents an effective treatment for metastatic and inoperable pheochromocytomas and paragangliomas. The median OS and PFS found in this study are comparable to those reported in previous studies employing PRRT. The efficacy and safety of PRRT provide support for including PRRT in the treatment algorithm for this patient group. Nevertheless, data on efficacy must be approached with caution, as they are primarily derived from retrospective studies, and cohorts are therefore not necessarily directly comparable. To establish the most appropriate treatment algorithm, including how treatment should be sequenced, the need for RCTs remains crucial.

## Figures and Tables

**Figure 1 cancers-16-01349-f001:**
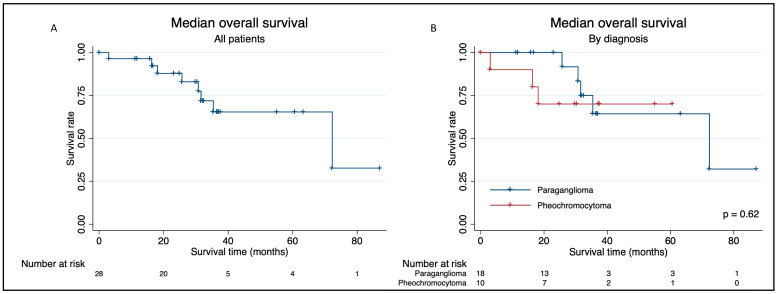
Kaplan–Meier survival plots displaying median overall survival after the first PRRT cycle for; (**A**) all patients; and (**B**) stratified by diagnosis (paraganglioma; pheochromocytoma).

**Figure 2 cancers-16-01349-f002:**
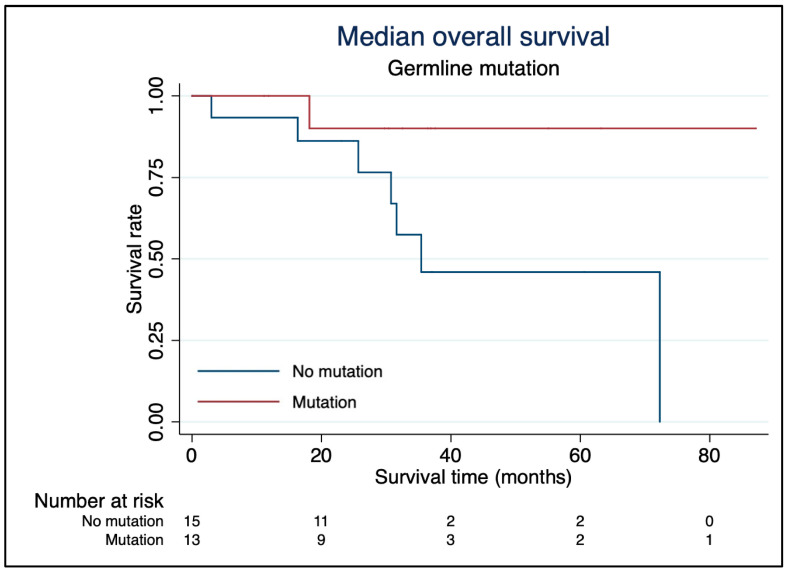
Kaplan–Meier survival plots displaying median overall survival after the first PRRT cycle stratified by germline mutations.

**Figure 3 cancers-16-01349-f003:**
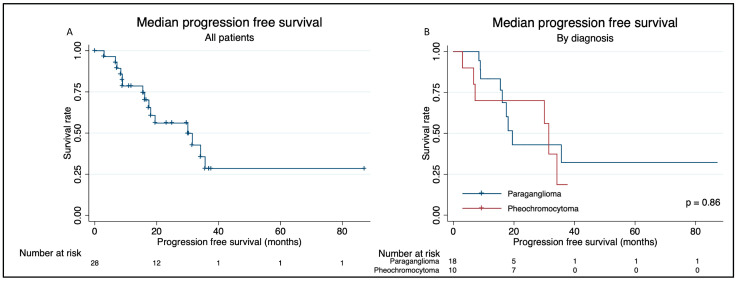
Kaplan–Meier survival plots displaying median progression-free survival for; (**A**) all patients; and (**B**) stratified by diagnosis (paraganglioma; pheochromocytoma).

**Table 1 cancers-16-01349-t001:** Demographics, tumour classification, and laboratory analyses. pCgA: plasma chromogranin A.

Characteristics	All (N = 28)	Pheochromocytoma (N = 10)	Paraganglioma (N = 18)	*p*
Gender, n (%)				0.80
Men	17 (61)	6 (60)	11 (61)	
Women	11 (39)	4 (36)	7 (41)	
Median age at diagnosis, years (range)	47 (15–75)	46 (19–75)	49 (15–72)	0.76
Men	47 (19–75)	31 (19–53)	50 (25–75)	0.056
Women	46 (15–65)	54 (34–63)	39 (15–65)	0.45
T-classification				0.97
Tx	2	1	1	
T1	6	2	4	
T2	8	3	5	
T3	12	4	8	
N-classification				0.18
Nx	5	0	5	
N0	9	4	5	
N1	14	6	8	
M-classification				0.46
M0	8	3	5	
M1a	1	1	0	
M1b	7	3	4	
M1c	12	3	9	
Median pCgA (range), pmol/L ^a^	254 (74–31,400)	254 (89–1780)	303 (74–31,400)	0.71
Median p-metanephrine (range), nmol/L ^a^	0.19 (0.01–41)	0.41 (0.01–15.1)	0.17 (0.01–41)	0.66
Median p-normetanephrine (range), nmol/L ^a^	2.5 (0.27–131)	2.82 (0.27–71.1)	1.25 (0.34–131)	0.42
Mutations	13	5	8	0.93

^a^ Data available for 23/28 patients.

**Table 2 cancers-16-01349-t002:** Demographics, tumour localisation, germline mutations and treatments. * Treatment after PRRT; all other: treatment prior to PRRT. M: male; F: female; PRRT: peptide receptor radionuclide therapy; PGL: paraganglioma; PCC: pheochromocytoma; LN: lymph nodes; MIBG: meta-iodo-benzyl-guanidine; TKI: tyrosine kinase inhibitor.

No.	Gender	Age at First PRRT	Primary Tumour	Primary Tumour Localisation	Metastases	Germline Mutation	Ki-67 (%)	Surgery	External Radio Therapy	MIBG Therapy	Chemo Therapy	TKI	PRRT Cycles
1	M	33	PGL	Chest	LN neck	None	5–15			x *	x		2
2	M	49	PGL	Paravertebral	Adrenal gland, LN neck	None	NA		x *	x	x *		4
3	M	76	PGL	Head, neck, chest wall	Liver, bones	None	10		x *			x *	6
4	M	25	PGL	Neck, glomus caroticus, abdomen		SDHB	4	x *					4
5	M	52	PGL	Pelvic		SDHB	20				x *		14
6	M	54	PCC		Bones	RET	1	x					4
7	F	33	PGL	Glomus jugularis		None	3	x	x				4
8	M	55	PCC			RET	0	x					4
9	M	58	PCC			RET	NA	x		x			4
10	F	65	PGL	Chest, left atrium		None	50		x		x		4
11	F	23	PGL	Retroperitoneum	Mediastinum, abdomen	None	1						4
12	M	26	PCC		Liver, bones	None	25						4
13	M	68	PGL	Bladder		None		x		x		x	6
14	M	55	PGL	Abdomen	Liver, lungs, bones	None	22		x			x	4
15	M	66	PCC		LN mediastinum and abdomen	None	NA						6
16	F	47	PCC			SDHB	20	x				x	4
17	F	47	PCC		Liver, bones, LN	NF1	NA		x	x		x	4
18	F	64	PCC		LN	None	NA			x			1
19	F	65	PCC			None	50						4
20	F	67	PGL	Glomus jugularis		SDHD	<1						4
21	F	49	PGL	Abdomen		SDHB	10			x			4
22	M	56	PGL			SDHB	3						4
23	M	57	PCC			None	2					x	4
24	M	75	PGL			None	10			x			4
25	F	67	PGL	Neck		None	17						4
26	M	41	PGL	Neck	Liver, bones, LN	SDHD	1						4
27	F	68	PGL			SDHB	20						4
28	M	52	PGL	Abdomen	Liver, bones, LN	SDHB	60		x		x		4

## Data Availability

Data are contained within the article.

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
