# Peer review of "Effects of Peptide Receptor Radiotherapy in Patients with Advanced Paraganglioma and Pheochromocytoma: A Nation-Wide Cohort Study"

_cancers, 2024, doi:10.3390/cancers16071349_

Round 1

Reviewer 1 Report

Comments and Suggestions for Authors

Skibsted Kornerup L et al. presented a nationwide cohort study examining the impact of peptide receptor radiotherapy in 28 patients diagnosed with metastatic paraganglioma and pheochromocytoma.

While the study addresses an important aspect of treatment for these rare diseases, there are several significant concerns that need to be addressed prior to publication.

Major concerns:

The title of the study should be modified to reflect its focus specifically on "metastatic" paraganglioma and pheochromocytoma.

The ethical statement provided is inadequate. Authors are required to adhere to the principles outlined in the Declaration of Helsinki. Given that this is a pharmacological study, informed consent is essential, even for retrospective studies involving a small number of patients. Additionally, how were the authors able to treat these patients without obtaining proper ethical authorization?

Minor concerns:

Line 201 contains a typographical error ("metanalysis") that needs correction.

In my view, the references should be enhanced by incorporating a minimum of 15-20 additional articles distributed across the introduction and discussion sections.

Comments on the Quality of English Language

English is fine. Authors should solve only small typos.

Author Response

See attachement 

Reviewer 2 Report

Comments and Suggestions for Authors

This manuscript presents a valuable retrospective study on the median overall survival (OS) and median progression-free survival (PFS) of Danish patients treated with Peptide Receptor Radionuclide Therapy (PRRT) for paraganglioma and pheochromocytoma. The study's strengths lie in its nation-wide scope and the inclusion of a relatively rare patient group. However, some areas could be improved to enhance the manuscript's clarity, impact, and scientific contribution.

Major Comments:

1. Since the study focuses solely on patients treated with PRRT, without a direct comparative group receiving  131I-MIBG, it inherently limits the ability to conclusively determine which treatment should be considered first-line based on the data presented. In addressing this point in your peer review report, you might suggest that the authors clarify the scope of their conclusions regarding treatment selection. Specifically, they could highlight that while the study adds valuable data on the outcomes of PRRT in this patient population, direct comparisons with 131I-MIBG treatments, ideally through a prospective randomized controlled trial or a well-designed observational study comparing both patient groups, are necessary to guide first-line therapy decisions. This clarification would ensure that readers understand the limitations of the study in the context of treatment selection and the current evidence gap that future research needs to address.

2. Addressing the uniqueness and specific contributions of the study, especially when it's not the first or largest of its kind, is crucial for articulating its value to the field. The authors may consider more clearly delineate the novel aspects or unique contributions of their research. This could include emphasizing any of the following:

Unique Population or Setting: Highlighting if their study population or setting (e.g., nationwide data) offers new insights not covered by the other studies.

Methodological Advancements: Discussing any methodological improvements or differences in how data were collected or analyzed compared to previous studies.

New Findings or Trends: Pointing out if the study uncovered any new trends, survival outcomes, or side effects associated with PRRT that were not previously documented.

Comprehensive Data Analysis: Underlining the comprehensiveness of the data analysis, especially if the study included a longer follow-up period, a larger variety of endpoints, or a more detailed analysis of patient outcomes.

Contextual Relevance: Explaining the relevance of the study's findings to current clinical practice or policy, especially if it addresses a gap not tackled by earlier research.

Articulating these points in the discussion section will help clarify the study's significance, ensuring readers understand its unique contribution to the existing body of knowledge on PRRT treatment for paraganglioma and pheochromocytoma.

Minor comments:

1. Clarity and Organization: The manuscript is well-structured, providing a logical flow from introduction to conclusion. Nonetheless, it could benefit from a more detailed discussion on the implications of the findings for clinical practice, especially considering the heterogeneity of treatment responses among patients with different genetic mutations.

2. Literature Context: While the study provides an important contribution to the existing body of literature, a more thorough comparison with international studies on PRRT's effectiveness could offer deeper insights into how these findings fit into the global context.

3. Methodological Details: The eligibility criteria and PRRT protocols are well-described, contributing to the manuscript's reproducibility. Still, a discussion on potential biases inherent in retrospective studies and how they might affect the results would be valuable.

Comments on the Quality of English Language

The manuscript is generally well-written and the quality of English language is good, allowing for clear communication of research findings and implications. However, to further enhance the readability and professional quality, it is recommended to conduct a thorough proofreading session to correct minor typographical and grammatical errors. Additionally, ensuring consistency in terminology and scientific nomenclature throughout the document will improve its overall clarity. A professional language editing service could also be considered to refine language use and ensure the manuscript meets the publication's standards.

Author Response

See attachement 
